# Grape Seed Waste Counteracts Aflatoxin B1 Toxicity in Piglet Mesenteric Lymph Nodes

**DOI:** 10.3390/toxins12120800

**Published:** 2020-12-15

**Authors:** Daniela Eliza Marin, Cristina Valeria Bulgaru, Cristian Andrei Anghel, Gina Cecilia Pistol, Madalina Ioana Dore, Mihai Laurentiu Palade, Ionelia Taranu

**Affiliations:** National Institute for Research and Development for Biology and Animal Nutrition, INCDBNA Balotesti, Calea Bucuresti nr 1, Balotesti, 077015 Ilfov, Romania; cristinavaleria11@yahoo.com (C.V.B.); andrei.anghel@ibna.ro (C.A.A.); gina.pistol@ibna.ro (G.C.P.); dore.madalina@ibna.ro (M.I.D.); mihai.palade@ibna.ro (M.L.P.); ionelia.taranu@ibna.ro (I.T.)

**Keywords:** aflatoxin B1, grape seed meal, mesenteric lymph nodes, piglets

## Abstract

Aflatoxin B1 (AFB1) is a mycotoxin that frequently contaminates cereals and cereal byproducts. This study investigates the effect of AFB1 on the mesenteric lymph nodes (MLNs) of piglets and evaluates if a diet containing grape seed meal (GSM) can counteract the negative effect of AFB1 on inflammation and oxidative stress. Twenty-four weaned piglets were fed the following diets: Control, AFB1 group (320 μg AFB1/kg feed), GSM group (8% GSM), and AFB1 + GSM group (8% GSM + 320 μg AFB1/kg feed) for 30 days. AFB1 has an important antioxidative effect by decreasing the activity of catalase (CAT), superoxide dismutase (SOD), and glutathione peroxidase (GPx) and total antioxidant status. As a result of the exposure to AFB1, an increase of MAP kinases, metalloproteinases, and cytokines, as effectors of an inflammatory response, were observed in the MLNs of intoxicated piglets. GSM induced a reduction of AFB1-induced oxidative stress by increasing the activity of GPx and SOD and by decreasing lipid peroxidation. GSM decreased the inflammatory markers increased by AFB1. These results represent an important and promising way to valorize this waste, which is rich in bioactive compounds, for decreasing AFB1 toxic effects in mesenteric lymph nodes.

## 1. Introduction

The gut is constantly exposed to potentially harmful contaminants from food or feed, such as mycotoxins [1]. Aflatoxin B1 is a mycotoxin produced by different species of fungi, especially *Aspergillus flavus* and *A. parasiticus* [2]. AFB1 is a potent carcinogen in humans and animals [3] and, for this reason, was classified in Group 1 of human carcinogens on the basis of toxicological data. Aflatoxins can contaminate different commodities such as cereals, nuts, dried fruits, and spices [4].

AFB1 is absorbed in the proximal part of the gut at high rates regardless of the species [5]. As the passage of the toxin across the intestinal barrier is very high, AFB1 can compromise the intestinal epithelium even before the absorption in the upper part of the gut. Indeed, it was shown that AFB1 induced inhibition of epithelial cell growth [6], an increase of apoptosis [6], and an increase of apoptotic markers (Bax and caspase-3) [7]. Additionally, AFB1 induced dysregulation of intestinal microbiota [8] and affected intestinal barrier function [9,10]. After absorption in the intestinal villi, the lipophilic compounds, such as AFB1, are first transported via the mesenteric lymphatic duct and then through the thoracic duct before entering the blood circulation [11]. Mesenteric lymph nodes (MSLs) are important for the proper functioning of the immune system of the gut, acting as filters for nutrients and microbial substances that enter through the lymph in the intestinal lamina propria [12]. MSLs are responsible for tolerance induction to food particles, but they also contribute to the prevention of the systemic inflammatory response that can result from the bypass of portal circulation [13].

AFB1 is a potent immunomodulator as the toxin has been shown to impair both the innate and acquired immune responses [14,15]. However, until now, little is known about the effect of AFB1 on gut immunity and, especially, mesenteric lymph nodes as local effectors of the immune response in the gut.

Grape seed meal (GSM) represents the residue left after grape seed oil extraction; this waste can be recycled in order to enrich the feed rations of farm animals [16,17]. GSM has a high content of bioactive compounds: polyunsaturated fatty acids (mainly linoleic acid, n-6), in addition to oleic acid (n-9), dietary fiber, phenolic acids, resveratrol, proanthocyanidins, and flavonoids [18,19]. In living organisms, the beneficial effects of bioactive plant compounds as counteracting agents for AFB1 toxicity have been investigated both in vivo and in vitro [20]. An in-vitro study performed on human hepatocytes has shown that an extract from palm kernel cake, rich in phenolics, has beneficial effects by upregulating the genes involved in the response to oxidative stress and downregulating the genes associated with inflammation and apoptosis in AFB1-exposed hepatocytes [21]. Additionally, in vivo studies have demonstrated that in rats, oxidized tea polyphenols can form a complex with AFB1 that leads to a decrease of AFB1 absorption and, implicitly, a reduction of its toxicity [22].

In particular, the bioactive compounds from grape seed were shown to have beneficial effects on intoxication with AFB1. In a recent study, the supplementation of an AFB1-contaminated diet with grape seed proanthocyanidin extract reduced AFB1 residue in the liver and significantly mitigated the negative effects caused by AFB1 in broiler chickens [23]. Through their alimentation, rich in cereals, especially maize, pigs are particularly exposed to mycotoxins. We have demonstrated that GSM addition into a diet contaminated with AFB1 ameliorates histological liver injuries, reduces the parameters associated with oxidative stress in piglets [24], and is able to change the microbiota composition already affected by AFB1 exposure [25]. The present study is a continuation of our previous studies and aims to investigate if AFB1 can affect mesenteric lymph nodes as the gate of toxic substances in intestinal lamina propria. Additionally, this study aims to evaluate if the administration of a diet containing grape seed meal can counteract the negative effects of AFB1 on inflammation and oxidative response.

## 2. Results

### 2.1. Effect of Grape Seed Meal and Aflatoxin B1 on Oxidative Damage in Mesenteric Lymph Nodes

AFB1 induces a significative decrease in the activity of the enzymes involved in the response to oxidative stress, with 27% for catalase (CAT; *p* = 0.0248), 22% for glutathione peroxidase (GPx; *p* = 0.0034), and 10.2% for superoxide dismutase (SOD; *p* < 0.0001), as compared with the control piglets (Figure 1). Additionally, exposure of piglets to 320 μg AFB1/kg feed, significantly decreased the total antioxidant status (−20.5%; *p* = 0.0001) associated with an increase of lipid peroxidation (+52.3%; *p* = 0.0001), as assessed by the TBARS method. When piglet diet included 8% of grape seed meal, no effect was observed on the activity of the enzymes involved in antioxidative defense, total antioxidant capacity (TAC), or lipid peroxidation. However, the concomitant administration of both grape seed meal and toxin resulted in a significant increase of GPx (by 119.3%; *p* = 0.0048), SOD (by 105.9%; *p* = 0.0046), and TAC (by 112%; *p* = 0.0180), as compared with the AFB1 group, while decreasing lipid peroxidation (by 12.3%; *p* = 0.018). GSM was not able to significantly increase CAT activity (*p* = 0.331, as compared with the AFB1 group).

### 2.2. Effect of Grape Seed Meal and Aflatoxin B1 on Inflammatory Cytokine Synthesis in Mesenteric Lymph Nodes

Piglets’ exposure to AFB1 contaminated diet was responsible for a a significant increase in the synthesis of IL-1β (*p* = 0.0068), IL-8 (*p* = 0.0010), and for a tendency of increase in the synthesis of IL-6, IFN-γ, and TNF-α (*p* > 0.05; Figure 2).

GSM diet did not affect the synthesis of inflammatory cytokines, the concentration of analyzed cytokines being very similar to the concentration measured in the mesenteric lymph nodes of the control group.

GSM administration to the piglets exposed to AFB1 (AFB1 + GSM diet) had an important anti-inflammatory effect, decreasing inflammatory cytokine synthesis towards the control level. Inclusion of GSM into the AFB1-contaminated diet (AFB1 + GSM group) was responsible for a decrease in the synthesis of IL-1β (8.57 ± 2.97 to 5.16 ± 1.75 μg/g tissue), IL-8 (34.7 ± 2.07 to 27.59 ± 3.07 μg/g tissue), IL-6 (81.9 ± 14.9 to 57.97 ± 9.88), and IFN-γ (8.13 ± 2.54 to 6.72 ± 1.47 μg/g tissue) towards the levels of the control group.

This decrease was significant for IL-1β (*p* = 0.0009), IL-6 (*p* = 0.044), and IL-8 (*p* = 0.0018) and not for IFN-γ (*p* = 0.371). GSM was not able to counteract the increase of TNF-α induced by the toxin (*p* = 0.389).

### 2.3. Effect of Grape Seed Meal and Aflatoxin B1 on Cell Signaling Pathways in Mesenteric Lymph Nodes

In order to validate the qPCR analysis and to demonstrate that GSM can restore the negative effects, such as inflammation and the oxidative response triggered by AFB1, we assessed by Western blot the expression of three proteins involved in NF-kB and Nrf2 signaling pathways: phospho-p38 MAPK, phospho-NF-kBp65, and Nrf2 (Figure 3A–C).

Feeding piglets with the diet contaminated with 320 μg AFB1/kg feed resulted in a significant nuclear increase in protein expression of both phosphorylated forms of NF-kBp65 (*p* = 0.049, Figure 3A) and p38 MAPK (*p* = 0.028, Figure 3B), while Nrf2 expression was not affected (*p* = 0.55, Figure 3C) in the AFB1 nuclear fraction when compared with control.

No significant alteration of the expression of these three proteins was noticed in the cytoplasm, although a slight increase of NF-kB expression in the AFB1 group was observed. As compared with the control, concomitant administration of both GSM and AFB1 resulted in a decrease of the expression of both phospho-p38 MAPK (4.86 ± 0.54 vs. 5.86 ± 0.59 A.U.) and phospho-NF-kBp65 (2.24 ± 0.28 vs. 2.19 ± 0.14 A.U.).

Additionally, the expression of Nrf2 increased toward the control level in the AFB1 + GSM group as compared with the control in both nucleus (1.64 ± 0.19 vs. 1.67 ± 0.69 A.U.) and cytoplasm (1.06 ± 0.11 vs. 1.17 ± 0.28 A.U.). The expression of the proteins involved in the inflammation signaling pathway decreased significantly in the AFB1 + GSM group as compared with the AFB1 group: −35% for phospho-p38 MAPK (*p* = 0.0017) and −27% for phospho-NF-kBp65 (*p* = 0.05), respectively.

### 2.4. Effect of Grape Seed Meal and Aflatoxin B1 on mRNA Gene Expression Involved in Inflammation

Exposure to AFB1-contaminated diet was responsible for a significant increase in the expression of the majority of the investigated genes. AFB1 contamination led to a highly significant increase in the expression of several proinflammatory markers: IL-8 (*p* = 0.001), IL-1β (*p* = 0.023), and IL-18 (0.07) (Table 1).

Looking at the gene expression encoding for signaling molecules with a key role in the inflammation pathway, a similar increase was observed for (i) transcription factors: c-Jun (*p* = 0.013), NF-kB p65 (*p* = 0.004), STAT3 (*p* = 0.011); (ii) MAP kinases: extracellular signal-regulated kinase ERK 1 (*p* = 0.039); (iii) metalloproteinase MMP2 (*p* = 0.001). When compared with the control diet, GSM alone had no effect on inflammatory markers or signaling molecules, with the exception of some genes, but resulted in a decrease in inflammatory response of the mesenteric lymph nodes, compared with the AFB1 group. In the AFB + GSM group, GSM was responsible for a significant decrease of the expression of ERK-1 (*p* = 0.006), c-Jun (*p* = 0.028), IL-18 (*p* = 0.002), and IFN-γ (*p* = 0.002), while the AFB1 × GSM effect for the other gene expressions was not significant (*p* > 0.05).

### 2.5. Correlations between Gene Expressions of Transcription Factors, MAP Kinases, Metalloproteinases, and Cytokines in Pigs Fed AFB1 and AFB1 + GSM Diets

In order to confirm the inflammatory effect of AFB1 and to better understand the mechanism involved in GSM action in counteracting the AFB1 effect, mathematical correlations were established between the expressions of transcription factors, MAP kinases, metalloproteinases, and cytokines in mesenteric lymph node samples derived from pigs fed AFB1 or AFB1 + GSM diets. As expected, the expression of all the inflammatory markers analyzed in the present paper was positively correlated (Figure 4).

Highly significant correlation was obtained between the expression of NF-kB, the key signaling molecule of the inflammation pathway, and the expression of (i) transcription factor c-Jun (*R*^2^ = 0.95); (ii) MAPK signaling molecules: ERK1 (*R*^2^ = 0.927), JNK1 (*R*^2^ = 0.808), JNK2 (*R*^2^ = 0.792); (iii) metalloproteinase MMP2 (*R*^2^ = 0.958); (iv) inflammatory cytokine IL-18 (*R*^2^ = 0.75). When mathematical correlations were performed between inflammatory gene expressions of lymph nodes derived from the AFB1 + GSM group and those of the AFB1 group (Figure 5), a highly significant correlation were observed between the expression of NF-kB and the expression of c-JUN (*R*^2^ = 0.965), p38 (*R*^2^ = 0.924), JNK 1 (*R*^2^ = 0.828), JNK 2 (*R*^2^ = 0.986), STAT 3 (*R*^2^ = 0.982), MMP 2 (*R*^2^ = 0.952), TNF-α (*R*^2^ = 0.817), IL-8 (*R*^2^ = 0.822), IL-18 (*R*^2^ = 0.872), and IFN-γ (*R*^2^ = 0.875), which clearly demonstrate the anti-inflammatory effect of GSM on the mesenteric lymph nodes of AFB1-intoxicated piglets.

## 3. Discussion

Mycotoxins are responsible for important decreases in productivity and health in the animal sector, and finding new solutions for the reduction of their toxic effects represent a continuous challenge for scientists. For some mycotoxins, such as aflatoxins, zearalenone, and ochratoxin, the use of high-affinity binders represents a promising solution as they can form stable complexes that cannot be adsorbed in the gut and are eliminated through feces, thus reducing the toxins’ systemic absorption [26,27].

According to Commission Regulation (EC) no 386/2009, mycotoxin adsorbents, which are able to mitigate the toxins’ effects through the reduction of intestinal absorption, were classified as a new functional group of feed additives. As stipulated in this regulation, the mycotoxin adsorbents should suppress or reduce the absorption, promote the excretion of mycotoxins and, at the same time, increase the quality of the feed, assuring the protection of public and animal health. At present, the use of mycotoxin adsorbents represents the best method for protecting animals against the harmful effects of contaminated feed [28,29].

From the mycotoxin adsorbents group, aluminosilicates are the most tested, but their use is limited due to their capacity to bind other molecules besides mycotoxins, such as vitamins and minerals that cannot be further absorbed in the gut [30]. Additionally, many binders have proven a high efficacy for mycotoxin adsorption using in-vitro abiotic systems, but these qualities were not confirmed by the in vivo studies [31].

Grape residues, as agroindustrial waste, are frequently discarded in open areas, representing a problem for the environment [32]. In order to avoid this, investigations have been done for their use for other purposes. Highlighting their chemical composition, rich in bioactive compounds, has led to their use as food/feed additives and pharmaceuticals [33]. For example, grape seed waste has been used in animal farm production in order to get functional foods, such as eggs or meat enriched in PUFAs or polyphenols [16,34].

The use of grape waste as binders of mycotoxins is of recent interest. Thus, the bioactive compounds from grape byproducts (grape pomace) were shown to efficiently adsorb AFB1 in abiotic systems [35]. Grape pomace was also efficient in reducing the urinary mycotoxin biomarker of AFB1 and zearalenone in an in-vivo trial, where piglets were fed a bolus contaminated with a mixture of mycotoxins (fumonisin B1, deoxynivalenol, zearalenone, AFB1, and ochratoxin A), through the reduction of gastrointestinal mycotoxin absorption [36].

As the information related to the effect of AFB1 on the lymphatic system and, in particular, the lymph nodes are scarce, the present study investigated the effect of AFB1 on the mesenteric lymph nodes of piglets after weaning. Additionally, we have investigated if a diet containing grape seed meal can counteract the negative effect of AFB1 on inflammation and oxidative response in piglets’ MSL.

Many studies have shown that oxidative stress represents one of the causes for AFB1-induced toxicity, which leads to the generation of reactive oxygen species (ROS), resulting in lipid, protein, and DNA damage and, consequently, cell injury [37].

Oxidative stress plays a negative role in chronic inflammatory diseases [38], and our previous data indicate that oxidative stress and inflammation are tightly correlated in animals exposed to mycotoxins [39,40].

In the present study, our results have shown that AFB1 has an important antioxidative effect by decreasing the activity of principal enzymes involved in the response to oxidative stress (CAT, GPx, SOD) and total antioxidant status while increasing lipid peroxidation, as compared to oxidative stress parameters in the MSL of control piglets. As a result of the exposure to AFB1, an increase of transcription factors, MAP kinases as signaling molecules, and metalloproteinases and cytokines as effectors of inflammatory response were observed in the mesenteric lymph nodes of the intoxicated piglets. According to the most recent data, AFB1 has a rather biphasic effect on the immune response, with a stimulatory effect in the first phase, followed by a suppressive action in the second phase [41]. Immuno-stimulation induced by AFB1 increases the synthesis of inflammatory markers and free radicals, which lead to chronic inflammation and cancer [42,43]. Anti-inflammatory activities of plant bioactive compounds have been reported in acute and chronic inflammation in animal models [44,45], and recent data have shown that they can be used for counteracting the toxic effect of AFB1 [24,46,47]. Our previous results have shown that GSM addition into AFB1-contaminated diet induced the amelioration of liver injuries and decreased inflammation and oxidative stress in the liver of intoxicated piglets by decreasing the MAPKs and NF-κB signaling pathways overexpressed by the AFB1 diet [24].

Although the immune system represents one of AFB1’s targets, few studies have investigated the potential beneficial role of bioactive compounds on the immune organs. For example, curcumin, a powerful plant antioxidant, decreases the weight of spleen and bursa of Fabricius, as well as the ratio of spleen/bursa, in broilers fed AFB1, towards similar values to the control group [48]. Additionally, in AFB1-intoxicated mice, lycopene, another plant bioactive compound, can alleviate AFB1-induced immunosuppression by increasing spleen weight, spleen coefficient, T-lymphocyte subsets, and IL-2, IFN-γ, and TNF-α gene expression in spleen [49]. Similarly, our results have shown that GSM was able to decrease proinflammatory cytokine gene expression and protein synthesis (IL-1β, IL-6, IL-8, IL-18, and IFN-γ), as well as the gene and protein expressions of important markers of signaling inflammation—ERK-1, c-Jun, phospho-p38 MAPK, phospho-NF-kBp65—as molecules involved in the NF-kB signaling pathway.

It was shown that phytocompounds can alleviate the immunosuppression induced by AFB1 in the spleen of mice through the inhibition of oxidative stress and mitochondria-mediated apoptosis [49]. Indeed, our results have shown that GSM induced a reduction of AFB1-induced oxidative stress by increasing the activity of enzymes involved in the response to oxidative stress (GPx and SOD) and decreasing lipid peroxidation.

The mechanism responsible for the beneficial effect of GSM in counteracting AFB1 toxicity is not clear. Recent studies have shown that grape pomace was able to efficiently bind AFB1, decreasing in this way the toxin’s absorption and increasing its excretion [35,36]. Likewise, our in vitro studies, performed on a panel of eight agroindustrial wastes, have shown that GSM was the most efficient binder of AFB1; these results suggest that this waste has a high capacity to bind AFB1 (Palade et al., submitted). The decrease of the toxic effects of AFB1 in the MSL of the AFB1 + GSM group, observed in our study, can be related to the capacity of GSM to adsorb AFB1 and increase its elimination.

Some hypotheses concerning the GSM way of action can be formulated based on previous literature studies. The first one, related to the high cellulose content (37.8%) of GSM [24], has a great potential to adsorb AFB1 by electrostatic attractions and hydrogen bonding, resulting in the formation of a mycotoxin monolayer on its surface [48]. The second one is based on the ability of polyphenols to form a complex with AFB1 mycotoxins. In a recent study, Lu et al. (2017) demonstrated that polyphenols from fermented tea can reduce AFB1-induced liver injury as they bind the toxin in a complex (C-AFB1 complex) and, consequently, inhibit AFB1 absorption and increase toxin elimination through feces [22]. Another interesting hypothesis is that of Ali Rajput et al., who considered that the protective effects of the bioactive compound proanthocyanidin, from grape seed, may be due to AFB1 biotransformation in the gut, which leads to the reduction of AFB1 absorption [23]. However, more studies are necessary to elucidate the interaction between grape waste and AFB1.

## 4. Conclusions

In conclusion, our study demonstrates that before being transported via the mesenteric lymphatic duct and entering the blood circulation, AFB1 exerts proinflammatory and pro-oxidative effects in the mesenteric lymph nodes of intoxicated piglets. Grape seed meal, a waste product generated after oil extraction, has been shown to have the capacity to reduce the inflammation and oxidative stress triggered by AFB1. These results represent an important and promising way to valorize this waste, rich in bioactive compounds, for decreasing AFB1 toxic effects on mesenteric lymph nodes.

## 5. Materials and Methods

### 5.1. Animals and Dietary Treatments

Twenty-four crossbred weaned piglets (TOPIGS-40), 4 weeks old, were randomly assigned to the experimental groups and fed the following treatments for 30 days: Control (fed a maize-soybean diet), AFB1 group (diet contaminated with 320 μg AFB1/kg feed), GSM group (diet with 8% grape seed meal included), and AFB1 + GSM group (basal diet with 8% GSM and 320 μg AFB1/kg feed), as previously described by Taranu et al. [50]. Each experimental group was represented by 6 piglets (2 pens/treatment and 3 pigs/pen). Piglets had access ad libitum to water and feed during the experimental period.

The experimental protocol was approved on 5 February 2020 by the Ethical Committee (no. 52/2014 of INCDBNA Balotesti) and the animal handling was done in accordance with EU Council Directive 98/58/EC and Romanian Law 206/2004.

Animals were slaughtered at the end of the experiment by exsanguination, and samples of the mesenteric lymph nodes were taken on ice and stored at −80 °C until the assessment of the immune and stress oxidative parameters.

### 5.2. Composition of the Grape Seed Meal

Dried grape seed meal (GSM), resulted after oil extraction, was provided by S.C. OLEOMET-S.R.L., Bucharest, Romania. Analyses consisting of chemical composition (fat, protein, ash, fibers), total and specific polyphenol content, as well as polyunsaturated fatty acids (PUFA) and antioxidant capacity, were performed for GSM characterization, as already described by Taranu et al. [51].

### 5.3. Toxin

Pure AFB1 (FERMENTEC, Jerusalem, Israel) was used to contaminate both AFB1 and GSM + AFB1 diets. Briefly, 50 mg toxin was dissolved in dimethyl sulfoxide and mixed with the control diet in order to achieve a final AFB1 concentration of 320 μg/kg diet. The final AFB1 concentration was confirmed by ELISA and UPLC analysis (320 ± 10.9 μg AFB1/kg diet). In the control diet, the AFB1 concentration was 2.4 ± 0.15 μg/kg diet. The diets were analyzed for contamination by other mycotoxins (ochratoxin A, fumonisins, deoxynivalenol, zearalenone); the levels found were under the EU limits for pigs.

### 5.4. Determination of Total Antioxidant Status

A total antioxidant capacity (TAC) assay has already been described by Marin et al. [39]. The method consists of the measurement of the absorption of 2,20-azinobis-[3-ethylbenzothiazoline-6-sulfonic acid cation (ABTS+)] in samples of mesenteric lymph nodes, and the results are expressed as mmol TEAC (trolox equivalent antioxidant capacity)/g tissue.

### 5.5. TBARS Assessment

Thiobarbituric acid reactive substances (TBARS) were measured in samples of frozen mesenteric lymph nodes, as already described [39]. The results are expressed as nmol/mg protein.

### 5.6. Enzyme Activity Assessment

The activities of superoxide dismutase (SOD), catalase (CAT), and glutathione peroxidase (GPx) were measured using Cayman kits (Cayman Chemical, Ann Arbor, MI, USA), according to the instructions provided by the manufacturer [52]. A Tecan microplate reader (SunRise, Vienna, Austria) was used for the measurement of the absorbance.

### 5.7. Cytokine Measurement

Cytokine concentration was assessed in mesenteric lymph nodes, as already described by Marin et al. [39]. Briefly, the homogenates from 1 g of frozen sample for each animal, in buffer with complete protease inhibitor cocktail, were used for the analyses of cytokine content by ELISA. Bradford assay was used for the analyses of total protein content. Monoclonal antiporcine antibodies for IL-1beta, IL-6, IL-8, and TNF alpha were used as capture antibodies, and biotinylated antiporcine IL-1beta, IL-6, IL-8, and TNF alpha were used as secondary antibodies (R&D Systems; Minneapolis, MN, USA). Results were presented as micrograms of cytokine/g tissue.

### 5.8. Extraction of Total RNA and cDNA Synthesis

RNA extraction followed by cDNA synthesis was carried out as described by Pistol et al. [53]. Briefly, a Qiagen RNeasy midi kit (QIAGEN GmbH, Hilden, Germany) was used for the extraction of total RNA from frozen mesenteric lymph node samples and then treated with ribonuclease inhibitors and purified on columns of silica gel. Concentration and quality were analyzed using a Nanodrop ND-1000 (Thermo Fischer Scientific, Waltham, MA, USA), and integrity was analyzed using agarose gel electrophoresis. Then, 100 ng of total RNA samples were used for the generation of cDNA using an M-MuLV Reverse Transcriptase Kit (Thermo Fischer Scientific, Waltham, MA, USA). A GeneQuerry™ Pig cDNA Evaluation Kit (ScienCell, Carlsbad, CA, USA) was used for the assessment of the absence of contamination with genomic DNA and the successful reverse transcription of tRNA to cDNA and cDNA quality.

### 5.9. Detection of Inflammatory and Signalling Gene Expression by qPCR Array

Real-time PCR was used to evaluate the expression of transcription factors: c-Jun, nuclear factor NF-kappa-B p65 (NF-kB p65), and signal transducer and activator of transcription 3 (STAT3); MAP kinases: extracellular signal-regulated kinase (ERK 1), c-Jun N-terminal kinase (JNK 1, JNK2) and p38, and metalloproteinase MMP2; cytokines: tumor necrosis factor (TNF alpha), interleukin beta (IL-1 beta), interferon gamma (IFN gamma), interleukin 8 (IL-8), and interleukin 18 (IL-18), as already described by Marin et al. [40]. The sequence of gene-specific primer pairs and the conditions used for the reactions have already been published in our previous papers [54,55,56]. Duplicates were performed for each gene, and melting curves were used for the confirmation of the formation of single PCR products. For all primers, negative controls were used, consisting of qPCR mix except for cDNA. Two reference genes—beta-actin (ACTB) and hypoxanthine-guanine phosphoribosyl-transferase (HGPRT)—were used for the relative quantification of gene expression changes, and the results were expressed as fold change, as compared with the control group (Fc), using the 2^(−ΔΔCT)^ method [57].

### 5.10. Western Blot Analysis

The protein expression level of three proteins involved in cell signaling—MAPK-p38, NF-kB phosphorylated form and Nrf2 (Nuclear factor erythroid 2-related factor 2)—were measured by Western blot. Cytoplasmic and nuclear fractions of tissue lysates were obtained using the protocol recommended by Thermo Fisher Scientific (Rockford, IL, USA NE-PER) for Nuclear and Cytoplasmic Extraction Reagent Kits, as described by Pistol et al. [53]. After assessment of protein concentration (Pierce BCA Protein Assay Kit, Thermo Fischer Scientific, Waltham, MA, USA), lymph nodes lysates, undiluted (cytoplasmic lysates) or ½ diluted (nuclear lysates), were separated on a 10% SDS-PAGE and transferred onto a 0.45-µm nitrocellulose membrane. After being blocked overnight with 5% nonfat dry milk, the membrane was incubated after washing with primary antibodies from Cell Signaling Technology (Beverly, MA, USA) for phospho-MAPK-p38 (rabbit antiporcine phospho-MAPK-p38), phospho-NF-kB/p65 (rabbit antiporcine phospho-NF-kB p65), β-actin (rabbit antiporcine β-actin), and Nrf2 (rabbit polyclonal antibody; Abbexa, Cambridge, UK) for 2 h at room temperature. Then, the membranes were incubated with a horseradish-peroxidase-conjugated antirabbit IgG antibody for 1 h (Cell Signaling Technology, Danvers, MA, USA). The immunoreactivity was assessed by using Clarity Western ECL Substrate (Bio-Rad, Hercules, CA, USA). A MicroChemi Imager (DNR Bio-Imaging Systems LTD, Neve Yamin, Israel) was used for developing immunoblotting images, and GelQuant software (DNR Bio-Imaging Systems LTD, Neve Yamin, Israel) was used for the evaluation of the level of protein expression. The results represent the ratio between the expression level of the protein of interest (p38 MAPK, NF-kB/p65, Nrf2) and β-actin.

### 5.11. Statistical Analyses

The differences for all the investigated parameters were analyzed using one-way ANOVA tests, followed by Fisher’s PSLD test.

## Figures and Tables

**Figure 1 toxins-12-00800-f001:**
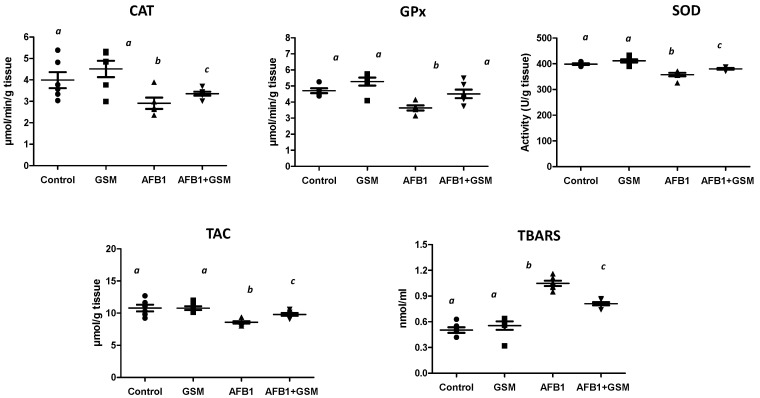
Effect of grape seed meal and AFB1 on oxidative damage in mesenteric lymph nodes. Data represent mean ± SD, *n* = 6. One-way ANOVA tests, followed by a Fisher’s PSLD test, were used for statistical data analyses. ^a–c^ indicate statistically significant differences between treatments (*p* < 0.05).

**Figure 2 toxins-12-00800-f002:**
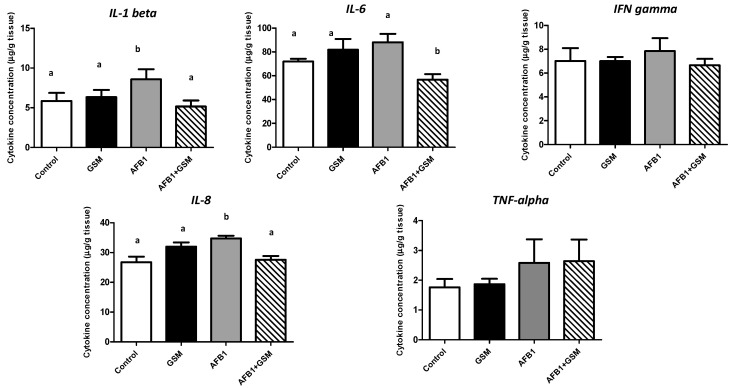
Effect of grape seed meal and AFB1 on inflammatory cytokine synthesis in mesenteric lymph nodes. Data represent means ± SD, *n* = 6. One-way ANOVA tests, followed by Fisher’s PSLD test, were used for statistical data analyses. ^a–c^ indicate statistically significant differences between treatments (*p* < 0.05).

**Figure 3 toxins-12-00800-f003:**
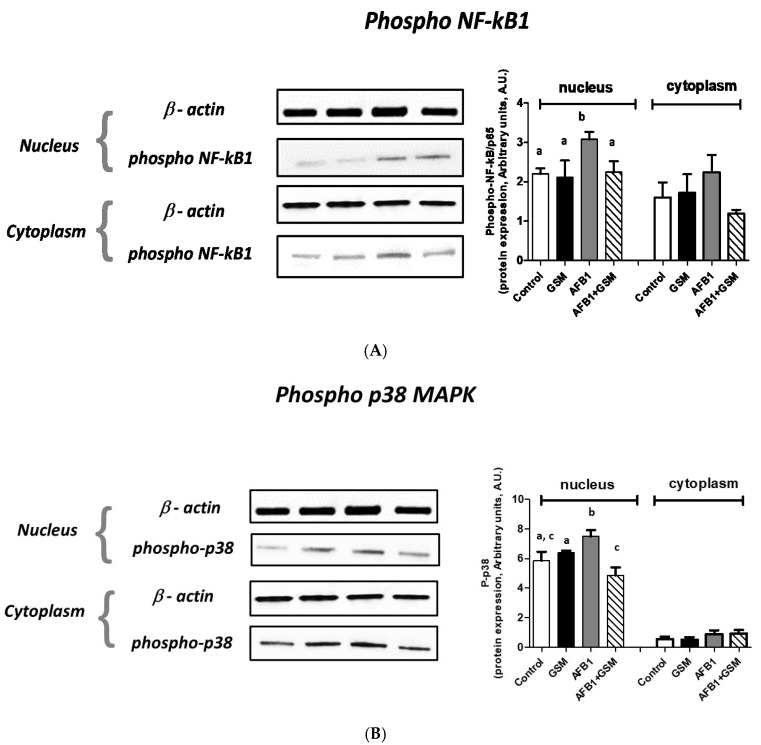
Effect of grape seed meal and aflatoxin B1 on cell signaling pathways in mesenteric lymph nodes: (**A**) phosphor-NF-kB1; (**B**) phosphor-p38 MAPK; (**C**) Nrf2. Data represent mean ± SD, *n* = 6. One-way ANOVA tests, followed by Fisher’s PSLD test, were used for statistical data analyses. ^a–c^ indicate statistically significant differences between treatments (*p* < 0.05).

**Figure 4 toxins-12-00800-f004:**
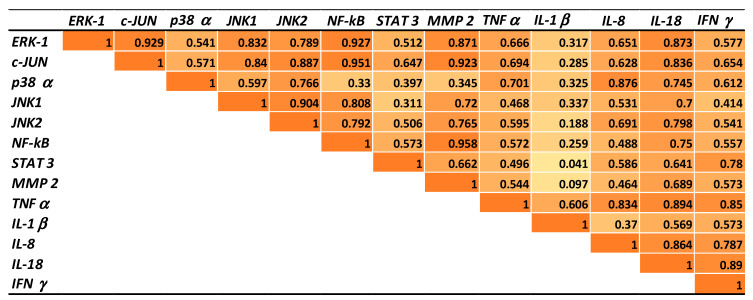
Gene expression correlations in lymph nodes of piglets exposed to AFB1 versus control.

**Figure 5 toxins-12-00800-f005:**
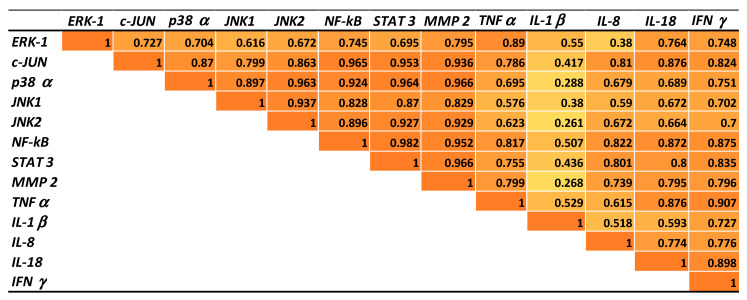
Gene expression correlations in lymph nodes of piglets exposed to AFB1+ GSM versus AFB1-exposed piglets.

**Table 1 toxins-12-00800-t001:** Effect of the exposure of piglets to aflatoxin B1 and/or aflatoxin B1 + grape seed meal on selected gene expression in mesenteric lymph nodes.

Gene Expression	Control	GSM	AFB1	GSM + AFB1	*p* Value	
					AFB1 Effect	AFB1 × GSM Effect
ERK-1	1.0 ± 0.73	3.53 ± 0.63	34.6 ± 0.63	1.96 ± 0.73	0.039	0.006
c-JUN	1.0 ± 0.28	0.94 ± 0.35	6.83 ± 0.62	1.07 ± 0.13	0.013	0.028
p38 a	1.0 ± 0.68	0.86 ± 0.03	2.21 ± 0.64	0.47 ± 0.24	0.108	0.782
JNK1	1.0 ± 0.59	0.56 ± 0.48	2.53 ± 0.57	0.78 ± 0.15	0.211	0.543
JNK2	1.0 ± 0.28	0.62 ± 0.51	3.12 ± 0.69	0.73 ± 0.25	0.170	0.297
NF-kB p65	1.0 ± 0.36	0.80 ± 0.66	7.97 ± 0.42	1.57 ± 0.08	0.004	0.112
STAT 3	1.0 ± 0.84	1.25 ± 0.62	17.4 ± 0.71	2.93 ± 0.17	0.011	0.681
MMP 2	1.0 ± 0.45	0.75 ± 0.58	12.5 ± 0.66	2.34 ± 0.10	0.001	0.168
TNF α	1.0 ± 0.10	0.95± 0.05	7.75 ± 0.49	3.22 ± 0.07	0.187	0.107
IL-1 β	1.0 ± 0.15	0.76 ± 0.34	15.3 ± 0.71	0.66 ± 0.10	0.023	0.073
IL-8	1.0 ± 0.49	0.94 ± 0.05	17.5 ± 0.37	0.87 ± 0.46	0.001	0.168
IL-18	1.0 ± 0.28	0.96 ± 0.61	26.8 ± 0.22	0.35 ± 0.14	0.076	0.002
IFN γ	1.0 ± 0.25	1.68 ± 0.82	14.2 ± 0.29	0.37 ± 0.14	0.326	0.002

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
