# Peer review of "Grape Seed Waste Counteracts Aflatoxin B1 Toxicity in Piglet Mesenteric Lymph Nodes"

_toxins, 2020, doi:10.3390/toxins12120800_

Round 1
Reviewer 1 Report
The article presented on the antitoxic potential of grape seed residues against AFB1 in mesenteric lymph nodes of piglets. And it concludes with an appreciable effect on inflammation and oxidative stress.
The article is well done and is interesting, although some modifications must be made:
- Eliminate the paragraphs between lines 161 and 173 and between lines 195 and 200, as they are superfluous.
-The paragraphs included between lines 174 and 194, must be moved from the discussion section to the introduction section.
- Justify why OTA, FUM, DON and ZEA are introduced into feed at concentrations lower than those legislated in the EU.
- The conclusions should state the reason why these effects occur. That undoubtedly will be due to the increased fecal excretion of AFB1. What would be easy to confirm by stool analysis.
Author Response
Reviewer 1
The article presented on the antitoxic potential of grape seed residues against AFB1 in mesenteric lymph nodes of piglets. And it concludes with an appreciable effect on inflammation and oxidative stress.
The article is well done and is interesting, although some modifications must be made:
Author response: Dear Reviewer, thank you for your appreciations concerning our work
- Eliminate the paragraphs between lines 161 and 173 and between lines 195 and 200, as they are superfluous.
Author response: Dear Reviewer, we agree that these phrases refers to general information, but we consider that they offer some basic information for a reader that is not very familiar with the oxidative stress, inflammation and the way that mycotoxins can affect these processes. We are kindly request your permission to keep this information in the discussion section.
-The paragraphs included between lines 174 and 194, must be moved from the discussion section to the introduction section.
Author response: Dear Reviewer, the mentioned paragraph is related to the first part of the discussion related to the mycotoxin adsorbents and the Commission Regulation (EC) no 386/2009. We believe that the transfer of this paragraph in the introduction section were we did not discuss about the mycotoxin adsorbents will be very confusing for the reader.
- Justify why OTA, FUM, DON and ZEA are introduced into feed at concentrations lower than those legislated in the EU.
Author response: Thank you for your comment. In the experiment only the diets for piglets belonging to the experimental groups 2 (AFB1) and 4 (AFB1+GSM) were artificially contaminated with AFB1 as described in the M7M section. No other mycotoxin was added in the diet. However, in order to be sure that the observed toxic effects were due to the AFB1 contamination and not to the contamination with other mycotoxin that could be presented in the diet due to a natural contamination the diets were analyzed for the presence of other frequently occurring mycotoxins (ochratoxin, fumonisin, deoxynivalenol and zearalenone). Our analyses indicated that the diets contain very low level or undetectable levels of these toxins (under the EU guidance level for piglets) that cannot affect the piglet health and that cannot be responsible for the toxic effects observed in our AFB1 intoxicated piglets.
- The conclusions should state the reason why these effects occur. That undoubtedly will be due to the increased fecal excretion of AFB1. What would be easy to confirm by stool analysis.
Author response: Thank you for your comment. As already stated in the introduction part: “After absorption in the intestinal villi, the lipophilic compounds, as AFB1, are first transported via the mesenteric lymphatic duct and then through the thoracic duct before entering the blood circulation (Kohan et al., 2011)”. So, in our opinion, the observed toxic effect on the mesenteric lymph nodes, are rather linked to the direct effect that the toxin exerts on the mesenteric lymph nodes during the absorption process and are less linked to the AFB1 that is not absorbed in the gut and eliminated by faces. Indeed, we can assume that GSM exerts its beneficial effect by adsorbing the AFB1 and reducing in this way its adsorption and the measurement of AFB1 in feces could represent a good way to investigate this effect. However, in this moment we cannot perform the analyze of the AFB1 content in the feces in our lab, but the method development is in progress, so we will be able to perform it in the near future.
According to your suggestion an explanatory phrase was added now to the conclusion.
“In conclusion, our study demonstrates that before being transported via the mesenteric lymphatic duct and entering the blood circulation AFB1 has a pro-inflammatory and pro-oxidative effects in mesenteric lymph nodes of intoxicated piglets”.

Reviewer 2 Report
Grape seed waste counteracts the aflatoxin B1 toxicity in piglets mesenteric lymph nodes
Toxins-1018598
This paper represents part of a study looking at the response of the gut mesenteric lymph nodes in piglets exposed to aflatoxin and whether this can be ameliorated by co-feeding with grape seed meal. This paper appears to be written from a lab that has published well on mycotoxins and grape seed meal in pig studies. Scientifically there is enough work completed to warrant publication of a manuscript. My initial thoughts about the paper were that broader research into the positive effects of the concomitant use of GSM is required. However, the authors appear to have published a lot of the required work and this is a further study using the same piglets as they have used in a previous study.
This is clearly a complementary study following on from previous work, which in my opinion is absolutely fine, but needs to be stated how this fits in with the previous research and that done by others. As there is no mention of this in the introduction or discussion, I believe that both of these sections need to be re-written to state clearly what they have done with this experimental set up already. Then why have they looked at these cells?
The conclusions and abstract are not justified by the current research as these results do not prove that GSM mitigates the effects of AFB1 toxicity, rather they show decreased toxicity in a specific set of cells.
The manuscript will also need to be thoroughly edited (re-written) for spelling grammar, sentence structure and consistency. The referencing has also been format in two different ways (author, year) in text and numbered in the bibliography.
The analysis for the graphs needs to be harmonised. Using the numbered significance is excellent. In figure 1 it is unclear what the “*”, “$” and “#” symbolise. Whilst showing 95 % CI is good, it is less meaningful with 6 replicates.
For all figures it would be good to clarify that treatments with different letters are significantly different. State the number of samples and statistical test applied. State what each of the graphs are in long form.
Figure 3. same as other figures, and have significance values on all panels.
Table 1. Are the statistics correct in this table?
Figures 4 and 5 are not clear, what do the numbers and colours represent. What is “Rela”? These need to be labelled correctly and also are there any better ways of showing this data?
Recently Rajput has shown grape seed phenolic extract -proanthocyanidin reduces toxicity in broilers. This may help the authors in addressing the mechanism of action of GSM.
For the above reasons I recommend that the manuscript should be reconsidered after major revision.
I have also listed a number of items that need to be corrected, but this is not excusive list.
Spell checking required and editing
Line 12. Glutathione has an “e”
Line 30, species.
Lien32. Change to AFB1 inhibits epithelial
Line 36. Consider changing to - after absorption in the intestinal villi, lipophilic compounds such as AFB1 are ..
Line 44 change to “was shown to impair both”
Line 60. Need to state somewhere that cereals are the major crops that are affected by aflatoxin. Which cereals are particularly important here?
Line 67. Better to name in full for first use in text even if done in abstract as done with GSM on line 48.
Line 69 Decreased
Line 69. How was total antioxidant status measured?
Line 74 and Figure 1. What is TCA?
Line 75. Is a 112 % decrease correct?
Line 81. Sentence needs to be re-written.
Line 85. As for Figure 1. Have significance values on all graphs.
Line 127. Table 1. Should use Greek letters for gene names where appropriate.
Line 147. R2 – should the “2” be superscript?
Could a heat map be a better method of showing the data?
Line 161. Rewrite sentence
Line 163. Adsorbent is probably a better word than binder which appears to be a more commercial name.
- what is Trolox?
- Change to “Fisher’s PLSD”
Author Response
Reviewer 2
This paper represents part of a study looking at the response of the gut mesenteric lymph nodes in piglets exposed to aflatoxin and whether this can be ameliorated by co-feeding with grape seed meal. This paper appears to be written from a lab that has published well on mycotoxins and grape seed meal in pig studies. Scientifically there is enough work completed to warrant publication of a manuscript. My initial thoughts about the paper were that broader research into the positive effects of the concomitant use of GSM is required. However, the authors appear to have published a lot of the required work and this is a further study using the same piglets as they have used in a previous study.
This is clearly a complementary study following on from previous work, which in my opinion is absolutely fine, but needs to be stated how this fits in with the previous research and that done by others. As there is no mention of this in the introduction or discussion, I believe that both of these sections need to be re-written to state clearly what they have done with this experimental set up already. Then why have they looked at these cells?
Author response: Thank you for your appreciations. Indeed, the present paper is a continuation of our previous work where we have described the beneficial effect of GSM on liver (Taranu et al 2020) and gut microbiota of the AFB1 intoxicated piglets (Grosu et al., 2019) and it was not in our intention to hide that this study belong to the same experiment used in our previous published papers.
We have taken into account your suggestions and we have modified the introduction and the discussion section accordingly in the new version of the manuscript.
The conclusions and abstract are not justified by the current research as these results do not prove that GSM mitigates the effects of AFB1 toxicity, rather they show decreased toxicity in a specific set of cells.
Author response: The conclusion as well as the abstract were changed according to the reviewer suggestions.
“In conclusion, our study demonstrates that before being transported via the mesenteric lymphatic duct and entering the blood circulation, AFB1 exerts a pro-inflammatory and pro-oxidative effects in mesenteric lymph nodes of intoxicated piglets. Grape seed meal, a waste product generated after oil extraction, has been shown to have the capacity to reduce the inflammation and the oxidative stress triggered by AFB1. These results represent an important and promising way to valorize this waste rich in bioactive compounds for decreasing the AFB1 toxic effects in mesenteric lymph nodes”
The manuscript will also need to be thoroughly edited (re-written) for spelling grammar, sentence structure and consistency. The referencing has also been format in two different ways (author, year) in text and numbered in the bibliography.
Author response: The manuscript was read and verified for grammar spelling, sentence structure and consistency. The referencing was verified and corrected.
The analysis for the graphs needs to be harmonised. Using the numbered significance is excellent. In figure 1 it is unclear what the “*”, “$” and “#” symbolise. Whilst showing 95 % CI is good, it is less meaningful with 6 replicates.
For all figures it would be good to clarify that treatments with different letters are significantly different. State the number of samples and statistical test applied. State what each of the graphs are in long form.
Figure 3. same as other figures, and have significance values on all panels.
Author response: Thank you for your comments. The presentation of the statistical significance was harmonized for all the figures. Also, we have modified all the figure legends as recommended by the reviewer, and the following information was added in the figure legend:
Data are mean ± SD, n=6. One-way ANOVA tests followed by a Fisher’s PSLD test were used for statistical data analyses. a,,b,c indicate statistically significant differences between treatments (P<0.05).
The significance values were already inserted in the text in order to not overcharge the figure 3.
Table 1. Are the statistics correct in this table?
Author response: We have double checked the statistics in this table as required by the reviewer. There were no mistakes in the statistical analyses and values.
Figures 4 and 5 are not clear, what do the numbers and colours represent. What is “Rela”? These need to be labelled correctly and also are there any better ways of showing this data?
Author response: We have considered that heat maps represent a good way to show the correlations in the lymph nodes between the different genes. The numbers represent the correlation value between two genes. In Figures 4 and 5 the colors indicate the correlation degree (yellow color very low correlation-dark orange high correlation). In the new version of the paper, according to the reviewer suggestions, the colors in the figures 4 and 5 were harmonized in order to be easily understood.
Recently Rajput has shown grape seed phenolic extract -proanthocyanidin reduces toxicity in broilers. This may help the authors in addressing the mechanism of action of GSM.
Author response: The suggested reference was added both in the Introduction and in the Discussion section.
For the above reasons I recommend that the manuscript should be reconsidered after major revision.
Author response: Dear Reviewer, we hope that with this new version of the manuscript we were able to answer to your comment and suggestions and that our joint effects have resulted in an improved version of the manuscript.
I have also listed a number of items that need to be corrected, but this is not excusive list.
Spell checking required and editing
Line 12. Glutathione has an “e”
Author response: Correction done
Line 30, species.
Author response: Correction done
Lien32. Change to AFB1 inhibits epithelial
Author response: Correction done
Line 36. Consider changing to - after absorption in the intestinal villi, lipophilic compounds such as AFB1 are ..
Author response: Correction done
Line 44 change to “was shown to impair both”
Author response: Correction done
Line 60. Need to state somewhere that cereals are the major crops that are affected by aflatoxin. Which cereals are particularly important here?
Author response: Correction done
A phrase concerning the cereals contamination with AFB1 was included in the Introduction. Contamination of maize with AFB1 represents a particular concern in term of piglets exposure. This is now clearly stated in the introduction section.
Line 67. Better to name in full for first use in text even if done in abstract as done with GSM on line 48.
Author response: Correction done
Line 69 Decreased
Author response: Correction done
Line 69. How was total antioxidant status measured?
Author response: The assessment of the total antioxidant capacity was presented in the M&M section. The method consists in the measurement of the absorption of 2,20-azinobis-(3-ethylbenzothiazoline- 6-sulfonic acid cation (ABTS+) in samples of mesenteric lymph nodes.
Line 74 and Figure 1. What is TCA?
Author response: It was an typo error. We referred at TAC (Total antioxidant capacity)
Line 75. Is a 112 % decrease correct?
Author response: It is decrease of 12.3%. sorry for the mistake
Line 81. Sentence needs to be re-written.
Author response: The sentence was reformulated: AFB1 was responsible for a significant increase of the synthesis of IL-1β (p= 0.0068) and IL-8 (0.0010) and for slight increase of the synthesis of IL-6, IFN-γ and TNF-α (p>0.05) (Figure 2).
Line 85. As for Figure 1. Have significance values on all graphs.
Author response: The significance values are already presented in the text and in our opinion, these values will overcharge the figures that contain already a lot of information.
Line 127. Table 1. Should use Greek letters for gene names where appropriate.
Author response: Correction done
Line 147. R2 – should the “2” be superscript?
Author response: Correction done
Could a heat map be a better method of showing the data?
Author response: We have considered that heat maps represent a good way to show the correlations in the lymph nodes between the different genes.
Line 161. Rewrite sentence
Author response: As suggested, the sentence was reformulated
Line 163. Adsorbent is probably a better word than binder which appears to be a more commercial name.
Author response: Correction done
what is Trolox?
Author response: Trolox (6-hydroxy-2,5,7,8-tetramethylchroman-2-carboxylic acid) is a water-soluble analog of vitamin E and it was used as standard for the determination of total antioxidant status. Our results are expressed in mmol TEAC (trolox equivalent antioxidant capacity)/g tissue
Change to “Fisher’s PLSD”
Author response: Correction done
Reviewer 3 Report
It is very interesting and good presentation. Mostly, the manuscript is scientifically sound, but some additional crucial assessment are required for better priority for publication.
- In terms of secreted cytokine production, author need to measure in the blood or serum. Tissue protein level is very low which is hardly representative of the AFB action. mRNA cytokine induction is very high. It 's really essential. Otherwise, author can measure secreted cytokine production in mesenteric lymphoid cells after cultivation.
- Prior mesentery, peyer's patch is really representative of the gut impacts. If the animal has PP, author can confirm the results in PP cells or gut lamina pripria lymphocytes.
- There is no mechanistic evaluation other than the correlation between cytokines and signaling molecule. Due to the animal experimental load, it is needed to verify the high relation using a simple cell culture model (lymphocytes or monocytes) by measuring the effects of the chemical inhibitors of signaling molecules (MAPK, NF-KB, or STAT3 inhibitors) on AFB-induced cytokine levels (protein or RNA) in the cells.
Author Response
Reviewer 3
It is very interesting and good presentation. Mostly, the manuscript is scientifically sound, but some additional crucial assessment is required for better priority for publication.
- In terms of secreted cytokine production, author need to measure in the blood or serum. Tissue protein level is very low which is hardly representative of the AFB action. mRNA cytokine induction is very high. It 's really essential. Otherwise, author can measure secreted cytokine production in mesenteric lymphoid cells after cultivation.
- Prior mesentery, peyer's patch is really representative of the gut impacts. If the animal has PP, author can confirm the results in PP cells or gut lamina pripria lymphocytes.
- There is no mechanistic evaluation other than the correlation between cytokines and signaling molecule. Due to the animal experimental load, it is needed to verify the high relation using a simple cell culture model (lymphocytes or monocytes) by measuring the effects of the chemical inhibitors of signaling molecules (MAPK, NF-KB, or STAT3 inhibitors) on AFB-induced cytokine levels (protein or RNA) in the cells.
Author response:
Dear Reviewer
Thank you for your comments. Undoubtedly our paper is far to address exhaustively the subject and the reviewer has right that a lot of work should be further perform. However, all the points raised by the reviewer concern studies that were not directly linked to our research and with the exception of the appreciation of our work as being interesting and good presented, there are no comments related to the quality of the work presented here but only comments related to additional experiments that should be performed. In our opinion, all the experiments suggested by the reviewers could stand alone as independent papers.
Our study has concerned the evaluation of the toxic effect of the AFB1 in mesenteric lymph nodes as an important gate for the access to the AFB1 into the organism as already stated in the introduction: the lipophilic molecules as AFB1 are first transported via the mesenteric lymphatic duct and then through the thoracic duct before entering the blood circulation. As already stated by other researchers (Watzl et al. 199) in the rat AFB1 is rapidly absorbed via the small intestinal tract that follows first-order kinetics (Ramos and Hernendez, 1996) and transferred almost completely into the mesenteric blood and transported also to the mesenteric lymph nodes (Kumagai, 1989). These authors suggest that these lymph nodes may be the organ tissue with the highest AFB1 exposure, and that was the reason for choosing to analyze the effect of the toxin on this tissue. We appreciate that it is more important to link the inflammatory gene expression with the inflammatory cytokine synthesis in the same organ rather than with the cytokine synthesis in the blood or serum.
Of course, the payer patch analyses could be of interest but also investigations concerning all the primary and secondary immune organs should be performed in order to have a complete image about the AFB1 effect on the immune system and the capacity of GSM to counteract this effect in piglets. Our present paper is a result of one year of work and the experiments requested by the reviewer on payer patches will request another additional year maybe more. Also, the study on the mesenteric lymph nodes cells requested by the reviewer should be performed on fresh cells. Our experiment was finished at the beginning of this year, so the lymph nodes were frozen transformed into a powder for the homogeneity of samples and the accuracy of analyses and consequently we need another in vivo trial in order to cultivated the MSL cells.
Referring to the mechanistic study, there are many in vitro studies that have shown that AFB1 is responsible for the induction of the inflammation in different cellular types including lymphocytes, so the experiment proposed by the reviewer concerning the effects of the chemical inhibitors of signaling molecules (MAPK, NF-KB, or STAT3 inhibitors) on AFB-induced cytokine levels (protein or RNA) in the cells will not bring any new information. In our opinion this mechanistic type of study is more suitable to assess the effect of a new compound/substance for that we are not sure that has an inflammatory effect.
Reviewer 4 Report
General Comments:
This study evaluated the ability of grape seed meal to offset the toxic effects of aflatoxin B1 in piglets. The authors sampled lymph nodes following dietary administration of treatments for 30 days and analyzed numerous dependent variables serving as indicators of inflammation and oxidative stress. This was a generally well-written manuscript, but I have outlined some issues below that must be addressed prior to acceptance for publication.
Were the piglets group-fed in a given pen? I have concerns that there were only 2 pen replicates per treatment. If the piglets were group fed, then that would make pen the experimental unit given the authors only 1 degree of freedom. Were there any treatment effects on average daily intake per pen or more importantly, per piglet? The authors need to provide more information on how the treatments were administered to the piglets. How can you assure the reader that each piglet received an equal quantity of treatment over the course of the 30-d trial?
Specific Comments:
L12 – Glutathione peroxidase
L16 – Important is misspelled
L25-26 – This sentence is verbatim repetition of L5-6. Suggest revision of one sentence or the other.
L67 – Should these abbreviations be defined separately from the abstract?
L79 – If the letters indicate significant difference between means, then what do the symbols mean? The figure caption should provide more information regarding the content of the figure. Should the figure stand alone
L150 and 158 – Can the authors provide information as to what the different colors are meant to indicate in these figures/tables?
L162 – change to “scientists”
L239-240 – This sentence needs to be rewritten. It is not clear what the authors are trying state here.
L251 – Change to “Grape seed meal, a waste product generated after oil extraction, has been shown to have the capacity … ”
L253 – Important is misspelled
L257 – Change to “24 weaned crossbred piglets …”
L261 – was this 8% of dry matter or as-fed basis?
L270 – There are numerous mesenteric lymph nodes. Did the authors sample the same nodes from each piglet? Were all nodes collected? What anatomical landmarks were used if not all nodes were collected to ensure a consistent sampling site across piglets?
L317 – Is this correct … liver powder? This is the first mention of liver being sampled from these animals. If correct, then this should be included in the sample collection section up around L269.
L372 – What about the effect of pen? If it is not the experimental unit, have the authors considered blocking on this to control for pen-to-pen variation?
Author Response
Reviewer 4
This study evaluated the ability of grape seed meal to offset the toxic effects of aflatoxin B1 in piglets. The authors sampled lymph nodes following dietary administration of treatments for 30 days and analyzed numerous dependent variables serving as indicators of inflammation and oxidative stress. This was a generally well-written manuscript, but I have outlined some issues below that must be addressed prior to acceptance for publication.
Were the piglets group-fed in a given pen? I have concerns that there were only 2 pen replicates per treatment. If the piglets were group fed, then that would make pen the experimental unit given the authors only 1 degree of freedom. Were there any treatment effects on average daily intake per pen or more importantly, per piglet? The authors need to provide more information on how the treatments were administered to the piglets. How can you assure the reader that each piglet received an equal quantity of treatment over the course of the 30-d trial?
Author response: Thank you for your general comments and observations and for the positive appreciation of our work. Each experimental group was represented by 6 piglets (2 pens/treatment and 3 pigs/pen). The number of piglets were sufficient in order to assure the statistical significance for this type of toxicological experiments that should limit at the maximum the number of the animals for ethics considerations.
The animals had a similar body weight at the beginning of the experiment and they were randomly distributed in the 4 experimental groups. The average daily gain per group was of 9.13 ± 0.03. The feed was administered in each pen and in a sufficient quantity in order to be assure ad libitum access of all the animals to the treatment. Administration of the treatment was responsible for a significative decrease (p=0.0013) of ADG in AFB1 intoxicated group (0.286 ± 0.046 vs 0.494 ± 0.056 in control group). GSM treatment has no effect on ADG as compared with the control (0.432 ± 0.025; p=0.3024) but there was a significant difference as compared with AFB1 group (p=0.0209). Treatment with both AFB1 and GSM resulted in a lower but still significant decrease of ADG as compared with the control (0.366±0.014; p=0.0327). There were no significant differences between the two pens belonging to different treatments (p= 0.91 inside Control group; p=0.90 inside AFB1 group; p=0.94 inside GSM group; p=0.96 inside AFB1+GSM group), the average daily gain of the animals belonging to the same group was very uniform and similar.
For these reasons, the pen was declared in the experimental design but only the effect of the treatment was taken into consideration.
Concerning your question related to the exposure of the animals, in the present paper, we have chosen the model of the intoxication through contaminated feed and not the intoxication by oral gavage because we want to reproduce farm conditions. Indeed, the gavage model can assure the exposure of each animal at a chosen dose, but this model has no relevance to agro-industrial practice and in consequence. Our animals were very homogenous at the beginning of the experiment and they had ad libitum access to feed and water all along the experiment. The feed intake has close values inside each treatment for each animal.
In a farm, the animals are intoxicated with mycotoxins through contaminated feed and all the animals that consume the contaminated feed will be intoxicated. Also, our experiment represents a model for a chronic intoxication, with a repeated exposure to the same contaminated feed and not an acute intoxication where the contaminations occurs accidental and only once. We believe that this type of experiments is superior to the experiment with oral gavage that are useful more in the toxico-kinetic trials. This model is used by our lab since many years and the results were published in peer review journals. Please find bellow some examples:
Marin DE, Braicu C, Dumitrescu G, Pistol GC, Cojocneanu R, Neagoe IB, Taranu I. MicroRNA profiling in kidney in pigs fed ochratoxin A contaminated diet. Ecotoxicol Environ Saf. 2019 Nov 30;184:109637. doi: 10.1016/j.ecoenv.2019.109637.
Taranu I, Marin DE, Palade M, Pistol GC, Chedea VS, Gras MA, Rotar C. Assessment of the efficacy of a grape seed waste in counteracting the changes induced by aflatoxin B1 contaminated diet on performance, plasma, liver and intestinal tissues of pigs after weaning. Toxicon. 2019 Mar 5;162:24-31. doi: 10.1016/j.toxicon.2019.02.020.
Grosu I.A., Pistol G.C, Taranu I., Marin D. E. The Impact of Dietary Grape Seed Meal on Healthy and Aflatoxin B1 Afflicted Microbiota of Pigs after Weaning Toxins 2019, 11(1), 25; doi.org/10.3390/toxins11010025 Marin DE, Pistol GC, Gras MA, Palade ML, Taranu I. Comparative effect of ochratoxin A on inflammation and oxidative stress parameters in gut and kidney of piglets. Regul Toxicol Pharmacol. 2017 Jul 29;89:224-231. doi: 10.1016/j.yrtph.2017.07.031.
Marin D.E., Motiu M. , Pistol G. C., Gras M.A. , Israel-Roming F., Calin L. , Stancu M. , Taranu I. 2016. Diet contaminated with ochratoxin A at the highest level allowed by EU recommendation disturbs liver metabolism in weaned piglets. World Mycotox J. 9 (4): 587 - 596
Pistol GC, Braicu C, Motiu M, Gras MA, Marin DE, Stancu M, Calin L, Israel-Roming F, Berindan-Neagoe I, Taranu I. 2015. Zearalenone mycotoxin affects immune mediators, MAPK signalling molecules, nuclear receptors and genome-wide gene expression in pig spleen. PLoS One. 2015 May 26;10(5):e0127503. doi: 10.1371/journal.pone.0127503. eCollection 2015.
And many others…
Specific Comments:
L12 – Glutathione peroxidase
Author response: Correction done
L16 – Important is misspelled
Author response: Correction done
L25-26 – This sentence is verbatim repetition of L5-6. Suggest revision of one sentence or the other.
Author response: Correction done
L67 – Should these abbreviations be defined separately from the abstract?
Author response: Correction done
L79 – If the letters indicate significant difference between means, then what do the symbols mean? The figure caption should provide more information regarding the content of the figure. Should the figure stand alone
Author response: In the fig 1 there was an error, the letters and the symbols indicate the same differences. We also have harmonized all the figures in term of the use of letters and symbols. In the new version of the paper we have kept only the letters.
L150 and 158 – Can the authors provide information as to what the different colors are meant to indicate in these figures/tables?
Author response: In Figures 4 and 5 the colors indicate the correlation degree (yellow color very low correlation-dark orange high correlation). In the new version of the paper, according to the reviewer suggestions, the colors in the figures 4 and 5 were harmonized in order to be easily understood.
L162 – change to “scientists”
Author response: Correction done
L239-240 – This sentence needs to be rewritten. It is not clear what the authors are trying state here.
Author response: The indicated sentence was reformulated in the new version of the manuscript.
L251 – Change to “Grape seed meal, a waste product generated after oil extraction, has been shown to have the capacity … ”
Author response: Correction done
L253 – Important is misspelled
Author response: Correction done
L257 – Change to “24 weaned crossbred piglets …”
Author response: Correction done
L261 – was this 8% of dry matter or as-fed basis?
Author response: GSM inclusion into feed was 8% as fed basis
L270 – There are numerous mesenteric lymph nodes. Did the authors sample the same nodes from each piglet? Were all nodes collected? What anatomical landmarks were used if not all nodes were collected to ensure a consistent sampling site across piglets?
Author response: As the reviewer also agree there is almost impossible to collect all the mesenteric lymph nodes because sometimes, they are very small and hardly visible. However, in order to assure a consistent sampling, we have collected a number of 20 superior mesenteric lymph nodes for each animal located near the transvers mesocolon. In order to assure the homogeneity of samples, all the collected lymph nodes for each animal were freeze-dried and grounded into a fine powder that were used for the analyses.
L317 – Is this correct … liver powder? This is the first mention of liver being sampled from these animals. If correct, then this should be included in the sample collection section up around L269.
Author response: Thank you for this remark. There was about samples of mesenteric lymph nodes powder, not liver powder. Sorry for this mistake.
L372 – What about the effect of pen? If it is not the experimental unit, have the authors considered blocking on this to control for pen-to-pen variation?
As already presented before there was no pen effect. As consequently, only the effect of the treatment was taken into consideration.
Round 2
Reviewer 1 Report
With the small changes made to the text, the article has improved and can be accepted for publication in toxins.
Reviewer 4 Report
No further comments